# The Nursehound *Scyliorhinus stellaris* Mitochondrial Genome—Phylogeny, Relationships among Scyliorhinidae and Variability in Waters of the Balearic Islands

**DOI:** 10.3390/ijms231810355

**Published:** 2022-09-08

**Authors:** Gaetano Catanese, Gabriel Morey, Francesc Verger, Antonio Maria Grau

**Affiliations:** 1Laboratorio de Investigaciones Marinas y Acuicultura (LIMIA-IRFAP)-Govern de les Illes Balears, Avinguda de Gabriel Roca 69, 07157 Port d’Andratx, Balearic Islands, Spain; 2INAGEA (UIB), Carretera de Valldemossa, km 7.5, 07122 Palma, Balearic Islands, Spain; 3Fundación Save the Med, Camí de Muntanya 7, 07141 Marratxí, Balearic Islands, Spain; 4TRAGSATEC, Passatge de Cala Figuera 6, 07009 Palma, Balearic Islands, Spain; 5Dirección General de Pesca i Medi Marí. Govern de les Illes Balears, calle Reina Constança, 4, 07006 Palma, Balearic Islands, Spain

**Keywords:** sharks, Nursehound, *Scyliorhinus*, mitogenome, phylogeny

## Abstract

The complete mitochondrial DNA sequence of the Nursehound *Scyliorhinus stellaris* has been determined for the first time and compared with congeneric species. The mitogenome sequence was 16,684 bp in length. The mitogenome is composed of 13 PCGs, 2 rRNAs, 22 transfer RNA genes and non-coding regions. The gene order of the newly sequenced mitogenome is analogous to the organization described in other vertebrate genomes. The typical conservative blocks in the control region were indicated. The phylogenetic analysis revealed a monophyletic origin of the Scyliorhininae subfamily, and within it, two subclades were identified. A significant divergence of *Scyliorhinus* spp. together with *Poroderna patherinum* in relation to the group of *Cephaloscyllium* spp. was observed, except for *Scyliorhinus torazame*, more related to this last cited clade. A hypothesis of a divergent evolution consequent to a selective pressure in different geographic areas, which lead to a global latitudinal diversity gradient, has been suggested to explain this phylogenetic reconstruction. However, convergent evolution on mitochondrial genes could also involve different species in some areas of the world.

## 1. Introduction

The Nursehound *Scyliorhinus stellaris* is an elasmobranch belonging to the Scyliorhinidae family, order Carcharhiniformes, with a medium size (up to 160 cm), appearing as a bycatch in a variety of fishing gears throughout its distribution range and entering the markets [1,2,3]. This shark is widespread in the whole Mediterranean, as well as in the coastal waters of the eastern Atlantic Ocean, from Morocco to the North Sea on the Scandinavian coasts [4]. It inhabits rocky and algal covered bottoms, at a depth range of 2–380 m, and feeds on mollusks, crustaceans, *Osteichthyes* and other demersal fishes [4].

Due to its large size, patchy distribution and suspected population declines, *S. stellaris* is currently listed by the IUCN as Vulnerable globally [5], whereas it is listed as Near Threatened in European and Mediterranean waters [6,7], and in the Balearic Islands, it has been listed as an “Endangered” species under criteria 1 (I, IV) in the Balearic Red List of Fishes [8].

The Nursehound could be locally extinct in some areas [9,10,11] as it has been documented across the Mediterranean Sea between the 1950s and 1990s [12,13]. However, the population recovery may be affected by low levels of interconnectivity between isolated populations around islands far from the continental coast [14].

From a taxonomic point of view, the family Scyliorhinidae is composed of 7 genera and about 50 species [4]. Within this family, Compagno et al. [15] described the monophyly of the subfamily Scyliorhininae, which includes the genera *Cephaloscyllium*, *Poroderma* and *Scyliorhinus*, in a hypothesis later supported by molecular studies [16,17]. Moreover, the taxonomic classification of some members of the genera *Cephaloscyllium*, including newly discovered species, has recently been revised based on morphological characteristics [18]. However, although further recent phylogenetic studies based on the mitochondrial genome have contributed to understanding the systematics and evolution of many species of sharks [19], the relationships between the genera within the Scyliorhininae subfamily have not yet been fully elucidated. *Cephaloscyllium* was hypothesized as the sister group of Scyliorhinus, based on clasper and skeletal characters [20], but a closer relationship between *Poroderma* and *Scyliorhinus* was purposed by Naylor et al. [21] when studying the *ND2* mitochondrial gene. Thus, some divergences in the phylogenetic relationships amongst its taxa have been observed between the morphological and molecular data. New data on the mitochondrial genomes of some species of the Scyliorhininae subfamily (*Cephaloscyllium umbratile* and *Poroderma pantherinum*) were also described to reveal their phylogenetic position [22,23], but the relationship of these species needs to be further studied.

Mitochondrial DNA (mtDNA) is commonly used in genetic studies due to its high abundance in the cell, high mutation rate and maternal inheritance [24]. The gene content of vertebrate mtDNA is a nearly identical set of 13 proteins, 22 tRNAs and 2 rRNAs, and a large non-coding region (control region) known to contain replication and transcription regulatory elements [25]. Mitochondrial genomes (mitogenomes) and their DNA molecular sequences (mtDNA) play important roles in evolutionary, genetic and ecological studies beside reliable datasets of mitogenomes which make molecular phylogenetic reconstruction very efficient [26].

The main objective of the present study was to sequence and characterize, for the first time, the complete mitogenome of the Nursehound *S. stellaris*. The gene content, organization, base composition and phylogenetic relationships, particularly with the Scyliorhininae subfamily species, including mitogenomes retrieved from GenBank, were studied. Finally, the diversity of the hypervariable mitochondrial control region of *S. stellaris* among individuals captured in the Balearic waters was analyzed.

## 2. Results and Discussion

### 2.1. Mitochondrial Genome Organization

The complete mitochondrial genomes of *S. stellaris* were 16,684 nt in length. The A + T content was 62.49% with the composition T = 31.53%, C = 23.63%, A = 30.95% and G = 13.89%. The full-length mitochondrial genome of *S. stellaris* has been deposited in the DDBJ/EMBL/GenBank databases and released under the accession number LC_723525. The organization and location of the different features in the genomes fit to the common vertebrate mitogenome model and other species of sharks [25,27]. Specifically, the mitogenome contained 13 protein-coding genes (PCGs), 2 ribosomal RNA (rRNA), 22 transfer RNA (tRNA) genes, the control region (D-loop) and very small nucleotide intergenic spacers (Figure 1; Table 1). The congeneric species *Scyliorhinus canicula* and *Scyliorhinus torazame* showed the same complete mitochondrial genome organization with a length of 16,697 nt (A + T = 62.01%) and 17,861 nt (A + T = 61.74%), respectively.

### 2.2. Ribosomal and Transfer RNA Genes

The 12S rRNA gene was 958 nucleotides long in *S. stellaris*, while it was 957 in *S. canicula* and *S. torazame*. With regard to the 16S rRNA gene, it was 1672, 1673 and 1669 nucleotides long in *S. stellaris*, *S. canicula* and *S. torazame*, respectively. A similar number of nucleotides were observed in other species of Scyliorhinidae: 955 nt in *P. pantherinum* as well as in *C. umbratile* and 957 nt in *C. fasciatum* for the 12S rRNA gene, while in relation to the 16S rRNA gene, they showed 1667, 1669 and 1672 nucleotides in *P. pantherinum*, *C. umbratile* and *C. fasciatum*, respectively. As in other vertebrates, in all Scyliorhinidae species, these rRNA genes were located between the tRNA-Phe and tRNA-Leu(UUR), being separated by the tRNA-Val.

The 22 tRNA genes in the *S. stellaris* mitogenome ranged in size from 67 to 75 nucleotides. Similar lengths were observed in all the species of the Scyliorhininae subfamily. They were interspersed between the rRNA and protein-coding genes. All the tRNAs could be folded into the typical cloverleaf secondary structure as determined by tRNAscan-SE software (Appendix A).

### 2.3. Noncoding Regions

Minor non-coding sequences varying from one to five nucleotides were also localized in the *S. stellaris* mitogenome between some coding regions (see Table 1, the positive values in the last column).

The L-strand replication origin (OL) was determined to be 51 bp in length in the *S. stellaris* and *S. canicula* mitogenomes but only 49 bp in length in *S. torazame*. As in most vertebrates, it was located within a cluster of five tRNA genes: tRNA-Trp (W, tryptophan), tRNA-Ala (A, alanine), tRNA-Asp (N, asparagine), OL, tRNA-Cys (C, cysteine) and tRNA-Tyr (Y, tyrosine), known as the WANCY region.

The major non-coding region, the control region or the D-loop, was located in the *S. stellaris* mitogenome between the tRNA-Pro and tRNA-Phe, and it was approximatively 1048 bp in length. The nucleotide content averaged 34.4% A, 33.8% T, 19.2% C and only 12.6% G for the L-strand. The same position of this region in the mitogenome of *S. canicula* and *S. torazame* was observed. However, these two species showed different lengths of the D-loop: 1050 nt for *S. canicula* and 1055 for *S. torazame*. In relation to the D-loop sequence of *S. canicula* and *S. torazame*, we observed 115 and 215 nucleotide variable sites, respectively. The length of the D-loop in the species of the Scyliorhininae subfamily was 1059 nt in P. patherinum and *C. umbratile*, while it was 1061 nt in *C. fasciatum*.

The control region includes the regulation and initiation sites of mitochondrial genome replication and transcription [28]. The typical domains with different conserved sequence regions, such as the Termination-Associated Sequence (TAS) at the 3′ end and the conserved sequence box (CSB D, CSB1, CSB2 and CSB3), were also suggested. In fact, aligning different blocks of conserved sequences of Scyliorhinus and other species of Carcharhiniformes, the putative TAS (TAS1, TAS2 and TAS3), CSB-D, CSB-1, CSB-2, CSB-3 and a pyrimidine tract could be identified (Figure 2). Stem-and-loop structures may be predicted using a comparative analysis among sequences, considering that they are generally flanked by 5′ A/T-rich regions in 3′, mainly situated in the middle of the region both in vertebrate and mollusks [29,30]. The CSBs are involved in the positioning of the RNA polymerase for transcription as well as for priming replication [31,32]. In this study, it is the first time that conserved blocks sequences are indicated broadly for Carcharhiniformes and more particularly for Scyliorhinidae.

#### Variability in Balearic Sea

The mitochondrial control region of *S. stellaris* was also used to investigate the genetic diversity from the samples captured in two different locations of the Balearic Sea. We observed 4 haplotypes from the 13 analyzed individuals that showed only three nucleotide polymorphic sites among their sequences. However, none of the haplotypes showed a clear association with any specific geographic location.

The sequences of the four haplotypes varied from 1048 to 1050 and they have been deposited in the DDBJ/EMBL/GenBank databases and released under the accession numbers: LC_723526- LC_723529. The haplotype diversity among the obtained sequence was Hd = 0.8077 and the nucleotide diversity was π = 0.00037. Although the number of samples in our study was lower, these values were similar to the overall results or slightly higher than those observed in the Balearic samples for *S. canicula* (Hd = 0.742, π = 0.0021), studying the genetic variability of the mtDNA COI sequences [33]. These discrepancies, in addition to being a consequence of studies on different species, are probably due to the type of marker used, demonstrating that an analysis of the D-loop region could be more sensitive and suitable in future population studies of *S. stellaris*.

### 2.4. Protein-Coding Genes

In *S. stellaris*, the majority of the PCGs were transcribed from the heavy (H) strand, except for the ND6 gene and 8 out of the 22 tRNA genes (tRNA-Gln, tRNA-Ala, tRNA-Asn, tRNA-Cys, tRNA-Tyr, tRNA-Ser, tRNA-Glu and tRNA-Pro), which were transcribed from the light (L) strand. All genes had a methionine (ATG) start codon except for the COI, which started with the GTG. The start GTG codon has also been reported at the beginning of the COI genes in other bony fishes [34,35,36,37]. The open reading frames ended with the TAA (ND1, ND2, COI, ATP8, ATP6, COIII, ND3, ND4L and ND6), AGA in COII and TAG (ND5). The two remaining genes (ND4 and cytB) showed an incomplete stop codon, “T”. The terminal T showed by some mtDNA genes is rather common to meet in vertebrates, where the TAA appears to be created via post-transcription polyadenylation [38].

Among the Scyliorhininae subfamily species, the same start and stop codons were shown in the comparison with *S. stellaris*, except for the termination of the genes: cytB, where *S. torazame* and *C. umbratile* showed the TAG; ND5, where all the other Scyliorhininae species showed the TAA; and ND6 gene, where *S. canicula* showed the TAG (Appendix A).

The nucleotide substitutions in the PCGs among the congeneric species varied from 10 (ATP8) to 127 (ND5) in the comparison of *S. stellaris* with *S. canicula* and from 24 (ATP8) to 249 (ND5) with *S. torazame*. In the comparison of the *S. stellaris* mitogenome with other species of the Scyliorhininae subfamily, the lowest overall number of substitutions was detected in the ATP8 gene, varying from 17 in *P. pantherinum* (10 synonymous and 7 nonsynonymous) to 24 in *C. umbratile* and *C. fasciatum* (16 synonymous and 5 nonsynonymous),while the highest varied in the ND5 gene from 222 nucleotide substitutions in *P. pantherinum* (180 synonymous and 41 nonsynonymous) to 240 in *C. fasciatum* (191 synonymous and 48 nonsynonymous) (Appendix A). The number of transitions (ts) and transversions (tv) varied from 9 (ATP8 of *S. canicula*) to 193 (ND5 of *P. pantherinum*) and from 1 (ATP8 and ND4L of *S. canicula*) to 61 (ND5 of *S. torazame*) (Appendix A).

In the comparisons of *S. stellaris* with the Scyliorhininae species, we observed that in all the coding genes, amino acid changes occurred. The changes in the amino acid composition produced by nonsynonymous nucleotide substitutions varied from 1 in ND4L (1.02% of the AA changes in relation to the number of AA in the gene) in the comparisons of *S. stellaris* with *S. canicula* to 42 in ND4 (3.7% AA changes) with *C. umbratile*. Anyway, the highest value of the percentage of AA changes was detected in ND6 (13.29%) in the comparison with *C. umbratile* (Figure 3; Appendix A).

As we can see in Figure 3A, some genes (ND2, ND3) showed clear differences in the trend of AA changes, mainly in the comparisons of *S. stellaris* with *S. canicula* and *P. pantherinum*. However, among all the studied species, these genes did not show a different trend in relation to the synonymous substitutions (Figure 3B). Therefore, we suggest that this disagreement could be the result of a divergent evolution consequent to a selective pressure caused by the different environmental conditions of the farthest geographic areas where each of the studied species usually lives. Adaptation mechanisms in heterogeneous habitats have recently been described for the widely distributed pelagic fish Sardinella longiceps, with a positive and diversifying selection involved in the oxidative phosphorylation complexes of the mitochondrial DNA regions [39]. However, the same authors also clarified that the effect of genetic drift in a population with a low effective population size (Ne) may leave similar signals as positive selection. For that reason, studies on the populations of the *S. stellaris* must be further investigated to understand if the effective size of the Nursehound populations in the Mediterranean Sea could be strongly reduced. Moreover, it can also be observed in Figure 3B that in the comparison with *S. stellaris*, the species *P. pantherinum* showed a lower percentage of synonymous substitutions than *S. torazame*, and a different trend in the ND6 gene.

### 2.5. Multigenes Phylogenies and Genetic Distance

As shown in Appendix A, excluding the outgroup, the genetic distances varied from 0.074 between *S. canicula* and *S. stellaris* to 0.2540 between Hemigaleus microstoma and *S. torazame*. Within the Scyliorhininae subfamily, the highest value was 0.1571 between *P. pantherinum* and *S. torazame*.

The most appropriate model GTR + G was selected by jModeltest software using the Akaike Information Criterion (AIC). The phylogenetic trees constructed using the two methods were consistent with high intermediate bootstrap values post probabilities. As expected, the separation of the Scyliorhininae subfamily clade from other Carchariniformes species was observed. However, within this group, two subclades were detected, supported by high values of bootstrap (Figure 4). Curiously, in these two subclades, the species were not separated by genera. In one clade, *C. umbratile* and *C. fasciatum* were grouped together as well as in the other clade *S. stellaris* and *S. canicula*. Nevertheless, these two clades also included *S. torazame* and *P. pantherinum* in the first and second group, respectively, and showed *S. torazame* paraphyletic with respect to the congeneric species *S. stellaris* and *S. canicula* (Figure 4).

Other authors have proposed *P. pantherinum* as the sister taxon of Scyliorhinus sp. [21,23], as well as *S. torazame* placed in a separate clade in relation to *S. canicula* and *S. stellaris* [16,40]. At the same time, the Scyliorhinus and Cephaloscyllium genera would fit with the hypothesis of a sister group based on morphological characters, as proposed by Soares and de Carvalho [20].

However, these suggested different hypotheses could be merged after a more careful observation of the descriptions of the Scyliorhininae subfamily species which have been grouped in this work. It seems evident that these species are distributed in different clades following an apparent geographical pattern. In fact, the *S. canicula* and *S. stellaris* species are common in the Mediterranean Sea, the North Sea and the Northeastern Atlantic Ocean; *P. pantherinum* is common in the Southeast Atlantic and South Africa; *S. torazame* is distributed in the Northwest Pacific (Japan, Korea and Taiwan); *C. umbratile* in the Western North Pacific (Japan Sea, East China Sea, Korea and Taiwan), possibly up to New Zealand; and finally, *C. fasciatum* in the Western Pacific Ocean (China to northwestern Australia) [41].

The geographical distance between the areas where each species usually lives would agree perfectly with the phylogenetic reconstruction obtained in this study. The evolutionary divergences accumulated among them in different environments show a clear pattern of distance that could be referred to the latitudinal diversity gradient relative to the global distribution of biodiversity. Conversely, *S. torazame* is closer to the Cephaloscyllium clade than to the congeneric *S. stellaris* and *S. canicula* species, showing the non-monophyly of the Scyliorhinus genus. This strongest relationship could instead be due to a convergent evolution phenomenon among the different genera that inhabit the same geographical area. Convergent evolution occurs when species occupy similar ecological niches and adapt in similar ways in response to similar selective pressures [42]. Some studies describe effects of multiple adaptive evolution (positive selection and convergent/parallel evolution) on mitochondrial genes. For instance, it has been proven that the positive selection can drive the survival of fishes in the deep-sea environment, mainly maximizing the use of limited energy sources [43], or it can mediate tolerance to physico-chemical stress in independent lineages of fish as a result of changes in highly conserved physiological pathways associated with essential mitochondrial processes [44].

However, more studies with more Scyliorhinidae species from different areas should be conducted to confirm this hypothesis.

## 3. Materials and Methods

### 3.1. Fish Sampling

Thirteen specimens were collected in 2021 from commercial catches (trammel nets and bottom trawl) in two different areas of the Balearic Sea (Menorca channel and Formentera; Northwestern Mediterranean Sea) (Spain). Those individuals were transferred to aquaria, where they were kept alive both in quarantine and exhibition tanks. The species identification was based in the morphologic and meristic features. A portion of the dorsal fin of each individual was excised and kept in absolute ethanol.

### 3.2. DNA Isolation, Amplification and Purification

Total genomic DNA was isolated from 30 mg of tissue using the Tissue Genomic DNA Extraction Kit (Macherey-Nagel, Duren, Germany), according to the manufacturer’s instructions.

For the amplification of the mitogenome, primer pairs were designed based on the partial sequences of *S. stellaris* available on GenBank, followed by several specific primer pairs designed from obtained sequences in this study (Appendix A). PCR reactions were performed in a total volume of 20 μL containing: 10 μL of Kapa Taq Ready mix (Sigma-Aldrich, Burlington, MA, USA), 8.2 μL of sterile water, 0.4 μL of each primer (stock 20 Mmol) and 1 μL of DNA at 50 ng/μL. The following conditions of PCR were used: an initial denaturation step at 95 °C for 3 min, followed by 35 cycles of denaturation at 95 °C for 30 s, annealing at 50–60 °C for 30 s and elongation at 72 °C for 5 min. All the PCR products were purified using a mi-Gel Extraction Kit (Metabion) and bidirectionally sequenced using the Sanger’s method at Secugen S.L. service (Madrid, Spain).

### 3.3. Sequence Analysis

The sequences were manually checked and assembled with Sequence Scanner (ThermoFisher, Waltham, MA, USA) and Bioedit [45] software. The final complete sequence annotation was performed using NCBI BLAST. The location of the two ribosomal RNAs (rRNAs) and the protein-coding genes were primarily identified using MITOS WebSever [46], Mitoannotator-MitoFish [47] and RNAfold web server. The boundaries of 13 PCGs and 2 ribosomal RNA (rRNA) genes were also determined by comparing them with the homologous genes of other *Scyliorhinus* species. The amino acid sequences of 13 PCGs were aligned by using Bioedit software under defaults settings. Transfer RNA (tRNA) genes were predicted using the programs tRNAscan-SE v1.21 [48]. Number of haplotypes, haplotype diversity (Hd) and nucleotide diversity (π) were computed with DNASP v. 6 [49].

### 3.4. Mitogenomes Comparison, Genetic Distance and Phylogenetic Analyses

The nucleotide sequences were aligned with MEGA 11 [50], using the MUSCLE algorithm, and the sequences of all protein-coding genes (PCGs) were later concatenated. The base composition, relative codon usage and the pairwise genetic distances (p-distance) of each gene were obtained using MEGA11 software.

The sequences were then analyzed with JModelTest v2.1.7 [51] using the Akaike Information Criterion (AIC; [52]) to select the appropriate model of evolution, as a guide to determine the best-fit maximum likelihood model. Both maximum likelihood (ML) and Bayesian Inference (BI) methods were adopted to reconstruct the phylogenetic relationships using MEGA11, with 1000 bootstrap replicates, and MrBayes v. 3.2 [53], respectively. In addition to the PCGs sequences of the subfamily Scyliorhininae species *Scyliorhinus stellaris*, *S. canicula* (Y16067), *S. torazame* (AP019520), *P. pantherinum* (MH321446), *C. umbratile* (KT003686) and *C. fasciatum* (MZ424309), the PCGs sequences of 20 species of Carcharhiniformes available in GenBank were used in the phylogenetic analyses (Figure 4). The sequence of *Isurus oxyrinchus* (order: Lamniformes, FJ572956) was also included in the analysis as an outgroup.

## 4. Conclusions

A complete mitogenome sequence in the *Scyliorhinus* genus has now been added to the list. In summary, the mitogenome of the Nursehound *Scyliorhinus stellaris* was found to have a conserved gene order comparable to Scyliorhinidae and in general to vertebrates. The control region was informative at an intra-specific level. This information will be valuable for performing population studies of this vulnerable species. No studies prior to this described the sequences of the control region and the conserved blocks in Scyliorhinidae. Our results suggested that the phylogenetic placement of *S. stellaris* was highly supported based on both the ML and BI analyses. The tree topologies corroborated the monophyly of the clade of Scyliorhininae and also its division in two subclades, more related to the natural habitat of the geographical areas where each species inhabits than the relationship between the genera. Finally, the mitochondrial genes could indicate signals of positive selections in some amino acid residues, a condition indicating adaptation to deep-sea environments.

The mitochondrial genome and its detailed analysis will provide a valuable genetic resource for further studies on population genetics, species identification, gene arrangement and the evolutionary and conservation studies of the *S. stellaris*.

## Figures and Tables

**Figure 1 ijms-23-10355-f001:**
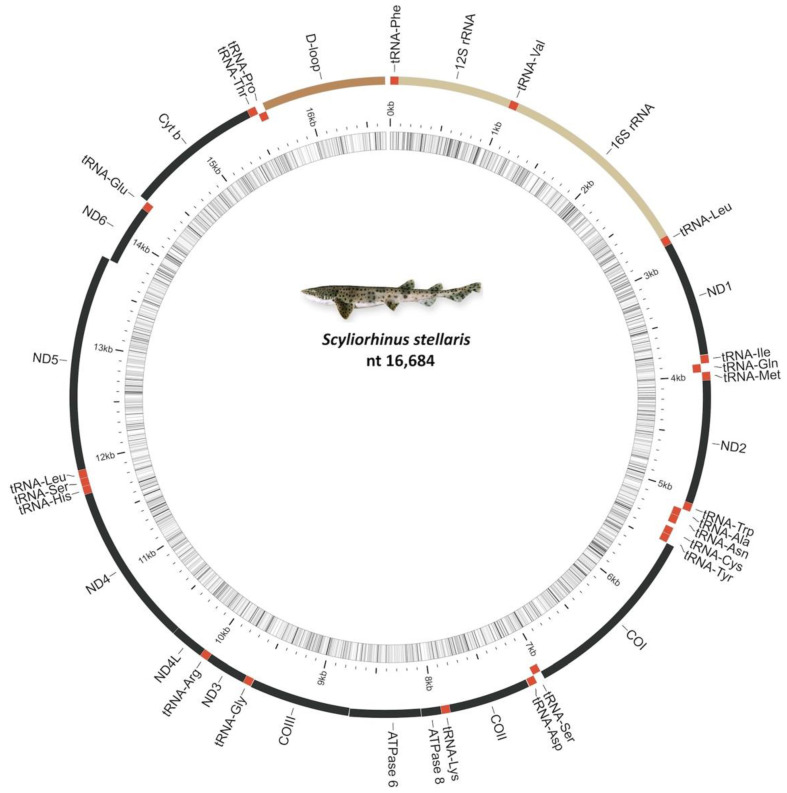
*Scyliorhinus stellaris* mitogenome drawn using the Mitoannotator Server (http://mitofish.aori.u-tokyo.ac.jp/annotation/input.html; accessed on 26 July 2022).

**Figure 2 ijms-23-10355-f002:**
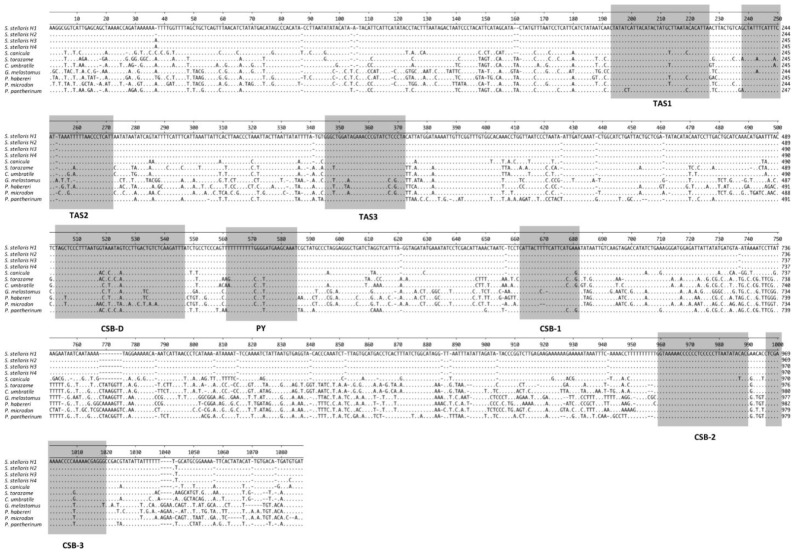
Alignment of the mtDNA control region of some species of Carcharhiniformes. The blocks TAS, CSB-D, CSB-1, CSB-2, CSB-3 and pyrimidine tract (PY) are shaded. Hyphens indicate gaps, and dots represent identity.

**Figure 3 ijms-23-10355-f003:**
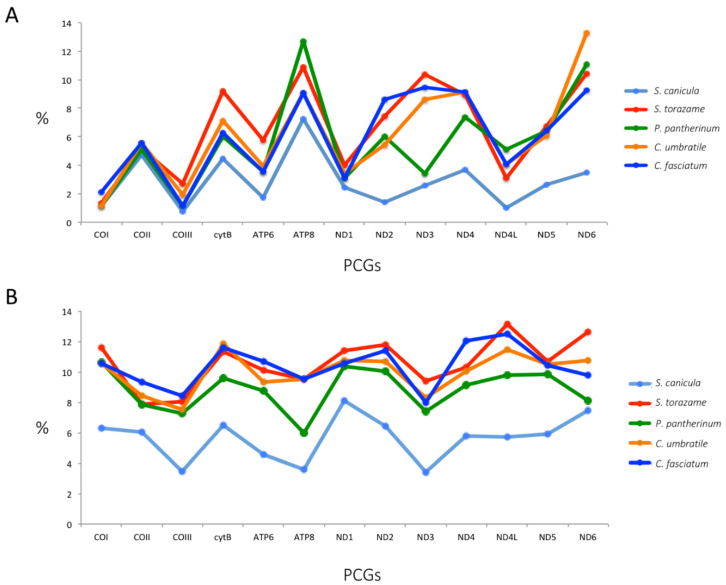
(**A**) Percentage of amino acids changes and (**B**) percentage of synonymous changes for each protein-coding gene in the comparison of *S. stellaris* with Scyliorhininae subfamily species.

**Figure 4 ijms-23-10355-f004:**
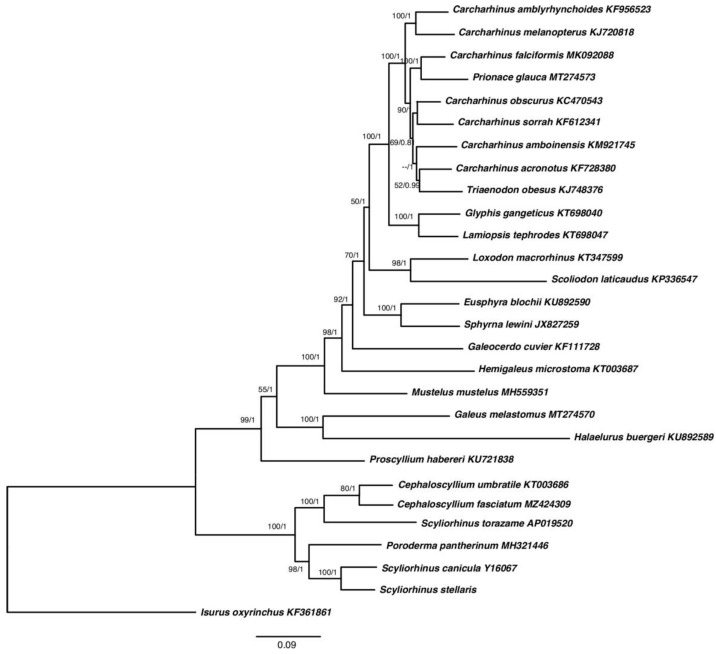
Phylogenetic tree based on concatenated protein-coding genes among Carchariniformes species. Maximum likelihood (ML) bootstraps and Bayesian (BI) posterior probability values are indicated above nodes.

**Table 1 ijms-23-10355-t001:** Location of features in the mitogenome of *Scyliorhinus stellaris*. L and H signify that the indicated gene is transcripted from L-strand or H-strand, respectively. Start and stop codons are indicated for protein-coding genes. The spacer column indicates the number of nucleotides downstream to the start of the next gene. The value is negative for genes with overlapping reading frames.

	Position		Strand	Length		Codons		Spacer
Gene Description	Start	Stop		nt	AA	Start	Stop	nt
**tRNA-F PHE (TTC)**	1	70	H	70				0
**rRNA12S**	71	1028	H	958				0
**tRNA-V VAL (GTA)**	1029	1100	H	72				0
**rRNA16S**	1101	2772	H	1672				0
**tRNA-L2 LEU (TTA)**	2773	2847	H	75				0
**ND1**	2848	3822	H	975	324	ATG	TAA	0
**tRNA-I ILE (ATC)**	3826	3895	H	70				3
**tRNA-Q GLN (CAA)**	3897	3968	L	72				1
**tRNA-M MET (ATG)**	3969	4038	H	70				0
**ND2**	4039	5085	H	1047	348	ATG	TAA	0
**tRNA-W TRP (TGA)**	5085	5153	H	69				−1
**tRNA-A ALA (GCA)**	5155	5223	L	69				1
**tRNA-N ASN (AAC)**	5224	5296	L	73				0
**OriL**	5294	5344	-	51				−3
**tRNA-C CYS (TGC)**	5334	5400	L	67				−11
**tRNA-Y TYR (TAC)**	5402	5471	L	70				1
**COI**	5473	7026	H	1554	517	GTG	TAA	1
**tRNA-S2 SER (TCA)**	7027	7097	L	71				0
**tRNA-D ASP (GAC)**	7098	7166	H	69				0
**COII**	7172	7867	H	696	231	ATG	AGA	5
**tRNA-K LYS (AAA)**	7860	7933	H	74				−8
**ATP8**	7935	8102	H	168	55	ATG	TAA	1
**ATP6**	8093	8776	H	684	227	ATG	TAA	−10
**COIII**	8776	9561	H	786	261	ATG	TAA	−1
**tRNA-G GLY (GGA)**	9564	9633	H	70				3
**ND3**	9634	9984	H	351	116	ATG	TAA	0
**tRNA-R ARG (CGA)**	9983	10,052	H	70				−2
**ND4L**	10,053	10,349	H	297	98	ATG	TAA	0
**ND4**	10,343	11,723	H	1381	460	ATG	T--	−7
**tRNA-H HIS (CAC)**	11,724	11,792	H	69				0
**tRNA-S1 SER (AGC)**	11,793	11,859	H	67				0
**tRNA-L1 LEU (CTA)**	11,860	11,931	H	72				0
**ND5**	11,932	13,761	H	1830	609	ATG	TAG	0
**ND6**	13,758	14,279	L	522	173	ATG	TAA	−4
**tRNA-E GLU (GAA)**	14,280	14,349	L	70				0
**CYTB**	14,352	15,495	H	1144	381	ATG	T--	2
**tRNA-T THR (ACA)**	15,496	15,567	H	72				0
**tRNA-P PRO (CCA)**	15,568	15,636	L	69				0
**DLOOP**	15,637	16,684	-	1048				0

## Data Availability

Molecular data have been deposited to GenBank with the following accession number: LC_723525-LC_723529.

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
