# Peer review of "The Nursehound Scyliorhinus stellaris Mitochondrial Genome—Phylogeny, Relationships among Scyliorhinidae and Variability in Waters of the Balearic Islands"

_ijms, 2022, doi:10.3390/ijms231810355_

Round 1
Reviewer 1 Report
A well written paper describing for the first time the complete sequence of the mitochondrial DNA of a nurse Scyliorhinus stellaris and compared to related species. The mitogenome sequence was 16,684 bp in length. The mitogenome consists of 13 PCG, 2 rRNA, 22 transfer RNA genes and non-coding regions. Typical conservative blocks in the Control Region are indicated. Phylogenetic analysis revealed a monophyletic origin of the Scyliorhininae subfamily, in which two subclades were identified. The authors showed that the mitogens Scyliorhinus stellaris have a gene sequence comparable to that of Scyliorhinidae and vertebrates. The obtained results may be valuable in conducting population studies of this species, and the mitochondrial genome and its detailed analysis will constitute a valuable genetic resource for further research on population genetics and species identification. I have no objections to the methodology, research conducted and the results described.
I suggest moving Figure 2 to the supplementary data.

Reviewer 2 Report
Review of The Nursehound Scyliorhinus stellaris mitochondrial genome. Phylogeny, relationships among Scyliorhinidae and variability in waters of the Balearic Islands by Catanese et al. for International Journal of Molecular Sciences
[I've also included a pdf in case this renders weird]
I will recommend this paper for minor revisions with some essential issues that need to be addressed. Most of this has to do with clarification and strange typos that I would assume would be addressed to some degree in the editorial stage.
Line 2 – Italicize binomial nomenclature in the title (and throughout)
Line 35 – weird spaces
Line 80 – consistency in numerical notation (e.g., 16,000 instead of 16.000)
Line 211 – confusing wording (does this mean that no Scyliorhininae gene is 100% conserved?) –whatever is being said here should be clarified.
Line 229 – if one is to open this question (selection versus drift in genic regions), maybe add some more analyses and figures like that of (or add a part b and part c to) figure 3 that restricts analysis to just nonsynonymous and synonymous substitutions.
The figures look great – methods are sound, appropriate, clearly written, and thorough. As far as I can tell the conclusions with respect to phylogeny and diversity of the group seem supported with the results as conveyed in the writing and figures. The results and discussion, however, can be unclear at times and entire manuscript needs to be revised and read over for grammar and clarity. Parts of it were difficult to read.
